# Elastic Properties of Alloyed Cementite M_3_X (M = Fe, Cr; X = C, B) Phases from First-Principle Calculations and CALPHAD Model

**DOI:** 10.3390/molecules29051022

**Published:** 2024-02-27

**Authors:** Yongxing Huang, Yang Lin, Guangchi Wang, Yehua Jiang, Xiaoyu Chong

**Affiliations:** Faculty of Material Science and Engineering, Kunming University of Science and Technology, Kunming 650093, China; 20212230113@stu.kust.edu.cn (Y.H.); ly101578@163.com (Y.L.); wguangchi@126.com (G.W.); jiangyehua@kust.edu.cn (Y.J.)

**Keywords:** cementite, first-principle calculation, calculation of phase diagrams (CALPHAD), elastic properties, brittleness–toughness

## Abstract

Fe-Cr-C-B wear-resistant steels are widely used as wear-resistant alloys in harsh environments. The M_3_X (M = Fe, Cr; X = C, B) cementite-type material is a commonly used strengthening phase in these alloys. This study investigated the mechanical properties of cementite (Fe, Cr)_3_(C, B) using the first-principle density functional theory. We constructed crystal structures of (Fe, Cr)_3_(C, B) with different concentrations of Cr and B. The bulk modulus, shear modulus, Young’s modulus, Poisson’s ratio, and hardness of the material were calculated, and a comprehensive mechanical property database based on CALPHAD modeling of the full composition was established. The optimal concentrations of the (Fe, Cr)_3_(C, B) phase were systematically evaluated across its entire composition range. The material exhibited the highest hardness, shear modulus, and Young’s modulus at Cr and B concentrations in the range of 70–95 at% and 40 at%, respectively, rendering it difficult to compress and relatively poor in machinability. When the B content exceeded 90 at%, and the Cr content was zero, the shear modulus and hardness were low, resulting in poor resistance to deformation, reduced stiffness, and ease of plastic processing. This study provides an effective alloying strategy for balancing the brittleness and toughness of (Fe, Cr)_3_(C, B) phases.

## 1. Introduction

Wear-resistant steel is typically alloyed with elements such as Si, Mn, Cr, Mo, W, V, Nb, Ti, and B [1,2] to improve its overall performance. In general, these alloying elements are introduced through two methods. In the first method, the solid solutions of these elements are incorporated in the steel matrix for solid-solution strengthening. In the second method, these elements are combined with other elements to create compound phases, primarily borides and carbides, which serve as second-phase strengthening agents [3,4,5,6]. Cementite is widely used to strengthen low-, medium-, and high-carbon steels. Specifically, cementite (*θ*-Fe_3_C) is the most prevalent and pivotal strengthening phase, which plays a significant role in the heat treatment and processing of steel [7].

In wear-resistant steel materials there are various types of carbides formed, and the carbides exhibit certain metal bonding characteristics resulting in the dissolution of other alloying elements through atomic substitution, forming complex multicomponent solid solutions. This is the main difference between the strengthening phase in steel and traditional compound phases [8]. The solubility of ca rbides is related to atomic radius, the number of outer electrons, and lattice type. Based on experimental statistics, the types of carbide strengthening phases in wear-resistant steel mainly include the following NaCl-type (B1-type) face-centered cubic lattice structure MC phases, such as VC, NbC, TaC, TiC, ZrC, HfC, etc. Non-metallic atoms in these phases often form vacancies, leading to a non-metal to metal ratio of less than 1. For example, the C content in VC ranges from 0.7 to 1, in NbC it ranges from 0.4 to 1, and in TiC it ranges from 0.5 to 1. Therefore, the chemical compositions of commonly existing VC and NbC in steel are VC_0.875_ (VgC,) and NbC_0.875_ (NbC,). Metal elements in MC phases can be completely mutually soluble, forming compounds like (V,Ti)C [8,9]. Simple hexagonal lattice MC and M_2_C phases include MoC, WC, Mo_2_C, and W_2_C, while complex hexagonal lattice M_7_C_3_ phases include Cr_7_C_3_ and Mn_7_C_3_. Mo_2_C and W_2_C can be completely soluble with each other. Cr_7_C_3_ can dissolve a considerable amount of Fe and Mn, and it can also dissolve certain amounts of W, Mo, V, and other elements [10]. Complex cubic lattice M_23_C_6_ phases include Cr_23_C_6_, Mn_23_C_6_, Fe_21_Mo_2_C_6_, and Fe_21_W_2_C_6_, etc. Cr_23_C_6_ can dissolve up to 25% Fe, and it can also dissolve some Mn, Mo, W, V, Ni, and other elements [11]. Complex cubic lattice M_6_C phases include Fe_3_Mo_3_C6 and Fe_3_W_3_C_6_, etc. In the M_6_C phase, W and Mo atoms can infinitely interchange with each other. Complex orthorhombic lattice M_3_C phases include Fe_3_C and Mn_3_C, etc., and they can be completely mutually soluble, forming (Fe,Mn)_3_C. Fe_3_C can dissolve a maximum of 28% Cr, 14% Mo, 2% W, or 3% V, forming alloy cementite [12].

It is known that the addition of chromium imparts such valuable properties as strength, hot hardness, and corrosion resistance. By dissolving in iron in the presence of carbon, chromium can form the carbides (FeCr)_3_C, (CrFe)_3_C_2_, (CrFe)_7_C_3_, and (CrFe)_4_C. The strength of the alloys is predominantly determined by the presence of the carbide phase, i.e., the cementite Fe_3_C, in which the solubility of chromium is as high as 18 at%. A further increase in the chromium content leads to the transformation (Fe,Cr)_3_C→(Cr,Fe)_7_C_3_→Cr_7_C_3_ [13]. Steels with an increased chromium content (6–32 at%) have a high wear resistance and the formation of the chromium carbide Cr_7_C_3_ plays a significant role in the improvement of their strength. The carbide Cr_7_C_3_ is thermodynamically stable. Recently, the metastable carbide Cr_3_C with a cementite structure (Fe_3_C-type) has been produced via rapid quenching. It is known that the chromium carbides exhibit unique properties, such as high hardness, chemical stability, and oxidation resistance [14].

White cast iron has been widely used as a wear-resistant material for a long time in many industrial applications. It is well known that the elastic properties of the alloy can be greatly affected by reinforced particles or precipitated phases. For white cast iron, the precipitated carbides usually refer to Fe_3_C. However, pure Fe_3_C is thermodynamically unstable. In practice, a small amount of Cr is added into ordinary white cast iron in order to stabilize Fe_3_C-type carbides, and as a result, the obtained carbides can be finally represented as (Fe, Cr)_3_C [15]. In current studies on the effect of boron on the wear resistance of Fe-Cr-B alloys containing different boron contents (0 wt%, 5 wt%, 7 wt% and 9 wt%), the boron element greatly improves the wear resistance of specimens as compared with that of an unreinforced specimen.

Cementite precipitates from either austenite or the liquid phase via eutectic reactions. The morphology and kinetics of cementite precipitation significantly affect the mechanical properties of steel. *θ*-Fe_3_C is a thermodynamically unstable metastable phase in Fe–C alloys. To obtain stable Fe_3_C-type carbides, alloying elements such as Cr and B are typically introduced into standard white cast iron. The resulting cementite alloy can be represented as *θ*-M_3_X (M = Fe, Cr; X = C, B) [16].

Lv [17,18] employed first-principles calculations to investigate the electronic structure, magnetic properties, and phase stability of Cr and Mn doped cementite alloys. The calculations revealed that the mixing enthalpies of Fe_8_Cr_4_C and Fe_4_Cr_8_C were negative. Furthermore, the cementite alloys exhibited enhanced chemical and mechanical stability when they were doped with Cr and Mn doping. Compared to *θ*-Fe_3_C, these alloys exhibited superior thermodynamic stability. In particular, compared to *θ*-Fe_3_C, (Fe,Cr)_3_C exhibited enhanced elasticity properties.

The inclusion of B as an alloying element in hard-phase tool steels offers several advantages, including enhanced thermal stability, hardness, and modulus of elasticity. Borides are formed through a direct reaction between Fe and B. Owing to the low solubility of B in a Fe lattice (500 ppm), only minimal quantities of B are required to create hard phases within significant bulk contents [19]. Furthermore, B improves the hardenability of the Fe matrix. Incorporating the alloying elements Cr and B further enhances the stability, modulus of elasticity, and hardness of carburizers, achieving a suitable balance between strength and toughness along with improved plasticity and machinability.

Cementite is a complex interstitial compound characterized by an orthorhombic crystal structure (space group Pnma, No. 62). As shown in Figure 1, the crystal cell contains 16 atoms, with metal atoms occupying two Wyckoff positions (4c and 8d) and non-metal atoms occupying one Wyckoff position (4c) [12]. Owing to the complexity of the structure and the inter-element interactions of carburized crystals, few studies have systematically investigated the influence of the composition of alloying elements on the mechanical properties of carburized hard phases. Therefore, this study aimed to quantitatively evaluate the mechanical properties of (Fe,Cr)_3_(C,B) alloys with respect to their compositional range using the CALPHAD method and first-principle calculations. The structure of θ-M_3_X (M = Fe, Cr; X = C, B) was constructed for various compositions. The positions of Fe and C atoms were replaced by Cr and B atoms, respectively, as shown in Figure 2.

## 2. Computational Methods

We employed first-principle calculations based on the pseudo-potential plane-wave method using the CASTEP quantum mechanics module [20]. The crystal wave functions were expanded using a plane-wave basis set, and the potential was represented using ultrasoft pseudo-potentials in reciprocal space. The exchange-correlation energy was computed within a generalized gradient approximation (GGA) framework using the Perdew–Burke–Ernzerh surface and solid (PBEsol) functional [21]. Fine precision settings were adopted to ensure the accuracy of the computational data and to maintain high efficiency. Periodic boundary conditions were applied and the number of plane waves was determined using the kinetic energy cutoff, which was set as 310 eV for all unit-cell models. The Brillouin zone was discretized using the Monkhorst–Pack method using an 8 × 6 × 9 k-point grid partition [22]. To ensure self-consistent convergence, the convergence thresholds for total energy, maximum stress, and maximum displacement were set as 1.0^−5^ eV/atom, 0.05 GPa, and 0.001, respectively, and a maximum iteration count of 500 was employed. Before each calculation, the crystal structure was geometrically optimized using the BFGS method to obtain the locally most stable structure. The valence electrons of Cr, Fe, C, and B are 3s^2^3p^6^3d^5^4s^1^4p, 3d^6^4s^2^, 2s^2^2p^2^, and 2s^2^2p^1^, respectively [23].

### 2.1. Elastic Properties at 0 K

The Birch–Murnaghan equation of state (EOS) [24,25] was adopted to fit the energy vs. volume (*E*–*V*) curve from the first-principles calculations at 0 K.
(1)$$E_{0}(V)=E(V)=E_{0}+\frac{9V_{0}B_{0}}{16}\left\{{\left[{\left(\frac{V_{0}}{V}\right)}^{\frac{2}{3}}-1\right]}^{3}B_{0}^{\prime }+{\left[{\left(\frac{V_{0}}{V}\right)}^{\frac{2}{3}}-1\right]}^{2}+{\left[6-4(\frac{V_{0}}{V})\right]}^{\frac{2}{3}}\right\}\;$$
*V*_0_ and *E*_0_ are the equilibrium volume per atom and the static total energy, respectively. *B*_0_ is the bulk modulus and $B_{0}^{\prime }={\left(\frac{\partial B}{\partial P}\right)}_{T}$.

The stress–strain method was used to calculate the elastic constants of single crystals and subsequently obtain the matrix of elastic coefficients of *θ*-M_3_X (M = Fe, Cr; X = C, B) single crystals. The second-order tensor of the elastic constants was based on the generalized Hooke’s law of elastic deformation [26]. Small strains were applied in different directions to optimize the positions of atoms in the crystal cell, the stress tensor of the deformed crystal cell was analyzed after deformation, and the elastic constants were obtained based on the stress–strain relationship:(2)$$\sigma_{i}=\overset{6}{\underset{j=1}{\sum }}C_{ij}\varepsilon_{j}$$
where *C_ij_*, *σ*_j_, and ε*_i_* are the elastic constant, stress tensor, and strain tensor, respectively. An orthorhombic crystal has nine independent crystal elastic constants: *C*_11_, *C*_22_, *C*_33_, *C*_12_, *C*_13_, *C*_23_, *C*_44_, *C*_55_, and *C*_66_. These constants can be determined by applying a small strain to the equilibrium lattice and computing the resultant change in the total energy of the lattice.
(3)$$\left\{\begin{array}{l} \sigma_{x}\\ \sigma_{y}\\ \sigma_{z}\\ \tau_{yz}\\ \tau_{zx}\\ \tau_{xy} \end{array}\right\}=\left[\begin{array}{cccccc} C_{11} & C_{12} & C_{13} & & & \\ & C_{22} & C_{23} & & & \\ & & C_{33} & & & \\ & & & C_{44} & & \\ & & & & C_{55} & \\ & & & & & C_{66} \end{array}\right]\left\{\begin{array}{l} \varepsilon_{x}\\ \varepsilon_{y}\\ \varepsilon_{z}\\ \gamma_{yz}\\ \gamma_{zx}\\ \gamma_{xy} \end{array}\right\}$$

By treating polycrystalline materials as aggregates of single crystals with random orientations, the isotropic polycrystalline elastic moduli can be computed as averages of the anisotropic single-crystal elastic constants. The theoretical lower and upper bounds of the true polycrystalline bulk (*B*) and shear moduli (*G*) are given by Reuss and Voigt, assuming uniform strain and stress throughout the polycrystal, respectively.
(4)$$B_{R}=\frac{1}{S_{11}+S_{22}+S_{33}+2\left(S_{12}+S_{23}+S_{13}\right)}$$
(5)$$B_{V}=\frac{C_{11}+C_{22}+C_{33}+2\left(C_{12}+C_{23}+C_{13}\right)}{9}$$
(6)$$G_{R}=\frac{15}{4\left(S_{11}+S_{22}+S_{33}-S_{12}-S_{23}-S_{13}\right)+3\left(S_{44}+S_{55}+S_{66}\right)}$$
(7)$$G_{V}=\frac{C_{11}+C_{22}+C_{33}-C_{12}-C_{23}-C_{13}}{15}+\frac{C_{44}+C_{55}+C_{66}}{5}\;$$
where *S_ij_* are the elastic compliances, and their values can be obtained by inverting the elastic constant matrix *S* = *C*^−1^. Based on the Voigt and Reuss models, *B_V_* (*B_R_*) and *G_V_* (*G_R_*) are the bulk and shear moduli, respectively [27]. *B* and *G,* based on the Hill model, can be calculated using their average values [28].
(8)$$B_{H}=\frac{\left(B_{R}+B_{V}\right)}{2}$$
(9)$$G_{H}=\frac{\left(G_{R}+G_{V}\right)}{2}$$

Young’s modulus (*E*) and Poisson’s ratio (*v*) were calculated as follows.
(10)$$E=\frac{9BG}{3B+G}\;$$
(11)$$\nu =\frac{3B-2G}{2(3B+G)}$$

*θ*-M_3_X is a highly crucial strengthening component in high boron wear-resistant alloys, characterized by its exceptionally high hardness. Hardness, which is a pivotal parameter for material wear resistance, was a central concern in this study. Hardness indicates the ability of a material to resist plastic deformation and failure. The subscripts *V* and *R* represent the Voigt and Reuss limits, respectively, and *σ* is Poisson’s ratio. In this study, we used the semiempirical equation of hardness proposed by Tian [29]. The following equation was defined to evaluate the hardness.
(12)$$H_{v}=0.92{\left(B/G\right)}^{1.137}G^{0.708}$$

In Tian’s article [29], a comparison of crystal structures, computed hardness, and experimental hardness for BC*_X_* was provided. This comparison encompassed various complex crystal structures, including orthorhombic structures such as BC*_3_* and BC_5_. Additionally, the calculated hardness for transition metal compounds such as FeC and FeC_2_ was also presented. These findings suggest that Tian’s computational model is applicable to orthorhombic structures such as (Fe,Cr)_3_(C,B). In Wang et al.’s research article, they investigated the relationship between the structural characteristics and mechanical behavior of multi-component iron-containing phases using Tian et al.’s proposed model [30]. Furthermore, Zhang et al. also utilized Tian’s model to calculate the hardness of (Fe, Cr)_7_C [31].

Anisotropic mechanical properties play an important role in material applications. The elastic anisotropy of a crystal can be estimated from independent elastic constants using anisotropic indices. In this study, the universal anisotropic index (*A^U^*) and percent anisotropic indices (*A_B_* and *A_G_*) were calculated using the following equations [32].
(13)$$\begin{array}{l} A^{U}=5\frac{G_{V}}{G}+\frac{B_{V}}{B}-6\geq 0\\ A_{B}=\frac{B_{V}-B_{R}}{B_{V}+B_{R}},A_{G}=\frac{G_{V}-G_{R}}{G_{V}+G_{R}} \end{array}$$

Here, *B_V_*, *B_R_*, *G_V_*_,_ and *G_R_* are the bulk and shear moduli estimated using the Voigt and Reuss methods, respectively. For isotropic structures, the values of the anisotropic indices are zero. Large discrepancies from zero indicate highly anisotropic mechanical properties. The elastic coefficient matrix of a single crystal of *θ*-M_3_X (M = Fe, Cr; X = C, B) phase was obtained using the stress–strain method based on the generalized Hooke’s law. Subsequently, the elastic modulus (*C_ij_*), including the bulk modulus (*B*) of *θ*-M_3_X, was obtained using the Voigt–Ruess–Hill approximation. The Born–Huang mechanical stability criterion was satisfied, indicating that the *θ*-M_3_X phases are all mechanically stable. For orthorhombic crystal systems, the mechanical stability criterion can be expressed as follows.
(14)$$\begin{array}{l} C_{i=j}>0,C_{11}+C_{22}-2C_{12}>0,\\ C_{11}+C_{33}-2C_{13}>0,C_{22}+C_{33}-2C_{23}>0,\\ C_{11}+C_{12}+C_{33}+2C_{12}+2C_{13}+2C_{23}>0 \end{array}$$

### 2.2. CALPHAD Modeling of Elastic Constants

The CALPHAD method is one of the few approaches capable of directly constructing a component–property relationship model from a multicomponent space. In 2010, Liu [33] modeled the variations in the mechanical properties of (Fe,Cr)_3_(C,B) solid solutions with concentration using the CALPHAD method. The CALPHAD method establishes a performance model for a multicomponent system. The general form of the model is shown below.
(15)Clm=Clmo+ΔClm
(16)Clmo=∑xioClmi
(17)ΔClm=∑i∑j>ixixj∑n=0Llmi,jnxi−xjn

Here, *i* and *j* denote pure elements. *^n^*$L_{lm}^{ij}$ is the binary interaction parameter of the elements *i* and *j. n* is the order. *x_i_ and x_j_* are the molar percentages of elements *i* and *j*, respectively. *C_lm_* is the elastic stiffness of the alloy. The variation in the elastic constants with concentration calculated by fitting the first-principles calculations was obtained using *^n^*$L_{lm}^{ij}$.

## 3. Results and Discussion

The total energies of the substitutional elements M (Fe,Cr) and X (C,B) in their respective ground-state structures were calculated as functions of volume at 0 K and zero pressure unless otherwise stated. The resulting EOS and lattice parameters are listed in Table 1.

Typically, the total energy of the relevant phase is calculated as a function of volume. The energy–volume relationship determines the equilibrium energy *E*(*V*_0_), equilibrium volume (*V*_0_), and bulk modulus (*B*_0_), as defined in Equation (1). As an example, Figure 3 shows the energy–volume relationship of the cementite M_3_X (M = Fe, Cr; X = C, B). The lattice parameters (a, b, and c) of the cementite at equilibrium volume were calculated by optimizing the interatomic forces and stresses in the unit cell.

Table 2 lists the elastic stiffness of M3X (M = Fe, Cr; X = C, B) along with the calculated elastic constants. Among the Fe3X (X = C, B) compounds, Fe3C exhibits the highest C11 value of 543 GPa, indicating the incompressibility of Fe3C. Compressing Fe3C under uniaxial stress along the [001] direction is more difficult. This is because elastic deformation induces a phase transformation from orthorhombic to monoclinic (space group P21/c), reducing its symmetry. Simultaneously, the number of three-dimensional covalent bonds increases which hardens the material. Fe8Cr4C4 exhibits the smallest value of C44 at 68 GPa. This indicates that, compared to other compounds, Fe8Cr4C4 is more susceptible to shear strains at the crystal plane (100). C11, C22, and C33 represent the ability of the crystals to resist axial strain along the [001], [010] and [001] crystallographic directions. C44, C55, and C66 indicate the ability to resist shear strain on the (100), (010) and (001) crystal faces, and C12 denotes resistance of the crystal against shear deformation along the [110] direction.

In the field of alloy design, this method has been successfully applied to many metal systems, e.g., the Nb single-bond Zr binary system [39], Ni-based alloys [40], Ti-Nb-Zr [41], the Zr-Nb-Mo ternary system [42], and even and intermetallic compounds. It is found that the crystal structure of Ti(Cu,Pt)_2_ is of the orthorhombic cell space group Amm2 (No. 38) with the structural prototype of VAu_2_. The resolved structure of Ti(Cu,Pt)_3_ is of the tetragonal AlPt_3_ type, belonging to the space group P4/mmm (No. 123) [43]. Ti(Cu,Pt)_3_ alloys of full composition were predicted and are in agreement with experimental data.

The recently developed performance-based CALPHAD modeling technique facilitates the rapid design of performance-oriented alloy compositions. This is founded on the ability of the technique to construct data models directly for material composition and properties [44]. Expanding the scope of alloy designing across the entire composition space opens up possibilities for obtaining various desired properties and combinations. Furthermore, it provides a rapid method for designing and optimizing alloys [40,45,46]. Figure 4 shows the elastic constants and CALPHAD model of the binary systems in the M_3_X (M = Fe, Cr; X = C, B) cementite. Table 3 lists the parameters fitted to the CALPHAD model.

The isotropic bulk modulus (*B*) and shear modulus (*G*) were determined. Generally, they cannot be calculated directly from *C_ij_*. Nevertheless, the values of the isotropic moduli can be confined within limits assessed in previous studies. Reuss obtained the lower bounds for all lattices, whereas Voigt obtained the upper bounds. Hill demonstrated that the averages proposed by Voigt and Reuss were limited and suggested that the actual effective moduli could be approximated to the arithmetic mean of the two bounds. We also calculated the Young’s modulus (*E*) and Poisson’s ratio (*v*) of the materials. These quantities are correlated with the bulk and shear moduli. Table 4 lists the calculated Voigt’s bulk modulus (*Bv*), Reuss bulk modulus (*B_R_*), effective bulk modulus (*B*), Voigt’s shear modulus (*Gv*), Reuss shear modulus (*G_R_*), effective shear modulus (*G*), Young’s modulus (*E*), and Poisson’s ratio (*v*) of M_3_X (M = Fe, Cr; X = C, B).

The model required for the calculation was generated using the sublattice point model. A finite number of constituent points were selected to construct the model, and the parameters reflecting the mechanical properties of the material, such as elastic constant *C_ij_*, bulk modulus *B*, shear modulus *G*, Young’s modulus *E*, Poisson’s ratio *v*, and hardness, were computed for each of the constituent points. Finally, the variation in the mechanical properties with the chemical composition was assessed across the entire compositional range by fitting the interaction parameters (Figure 5).

Parameters such as *B*, *G*, *E*, and Poisson’s ratio are crucial indices for evaluating the mechanical properties of the (Fe,Cr)_3_(C,B) phase in high boron anti-wear alloys. The bulk modulus characterizes the capacity of a material to resist volume changes. It serves as the evaluation standard for the average valence bond strength of the material. At the macroscopic scale, bulk modulus reflects the external homogeneity of the compression resistance against a hydrostatic pressure. The higher the resistance to deformation, the stronger the material, and consequently, it is more challenging to compress. Fe_3_C exhibited the largest *B* value of 320 GPa, while Fe_3_B has the smallest *B* value of 120 Gpa. The bulk modulus decreased gradually with increasing concentrations of B and Cr. Therefore, the capacity of the material to resist volumetric deformation decreased gradually with the increase in the concentrations.

The shear modulus is a measure of the ability of a material to resist shear stress. The higher the shear modulus, the stronger the resistance against shear strain. Hence, the modulus is closely related to hardness. The shear modulus *G* and hardness *Hv* increased with increasing B and Cr concentrations. That is, the resistance to shear strain increased, indicating that the plastic processing of the material became more difficult. Young’s modulus (E) is expressed as the extent of linear compression of the (Fe,Cr)_3_(C,B) phase. It reflects the ability of the material to resist positive strain. To a certain extent, it can also reflect the stiffness of the material. The higher the E, the higher the stiffness of the material. Cr_3_C exhibited the highest Young’s modulus and stiffness of 402 GPa and 18.1 Gpa, respectively (Table 4), indicating that Cr_3_C may be the stiffest material in the (Fe,Cr)_3_(C,B) phase. Poisson’s ratio (ν) is an essential parameter in material engineering. It is defined as the ratio of the strain perpendicular to the direction of the applied stress to the strain in the direction of the stress, when a material is subjected to unidirectional stress. Generally, Poisson’s ratio is a measure of both the elastic properties of a material and the stability of the material under the shear stress. It typically ranges between −1 and 0.5. The smaller the Poisson’s ratio, the higher the stability of the material.

The toughness and brittleness of a material can be assessed through the *B*/*G* ratio. According to Pugh’s standard, a B/G ratio of 1.75 is the threshold between brittle and ductile materials. A compound can be categorized as tough and brittle when *B*/*G* > 1.75 (ν > 0.26) and *B*/*G* < 1.75 (ν < 0.26), respectively. The calculated Poisson’s ratios of the materials were below 0.26, except for Fe_3_B, Cr_3_C, Fe_12_C_2_B_2_, Cr_12_CB_3_, Cr_12_C_3_B, and Fe_4_Cr_8_B. This indicates that the former materials are brittle and easily deformable under an external force. By contrast, the remaining compounds demonstrate good plasticity.

The elastic anisotropy index (*A^U^*) indicates the extent of variations in the mechanical properties of the material in different directions (Table 4). *A^U^* = 0 indicates that the material is isotropic. The greater the deviation of AU from 0, the higher the degree of elastic anisotropy in the material. As evident from Table 1, the total anisotropy index of Fe_8_Cr_4_C_4_ has the largest deviation from 0, whereas that of Fe_3_C has the second largest deviation from 0. This indicates that the mechanical properties of the two compounds are highly anisotropic. The anisotropy index of Cr_3_B (0.01) has the smallest deviation from zero, indicating that its mechanical anisotropy is the weakest.

Tian et al. proposed a model to predict the hardness of polycrystalline materials and bulk metallic glasses based on the Pugh’s modulus ratio and the shear modulus (G), where Hv denotes the hardness. The results for compounds are presented in Table 4. Umemoto et al. [54] reported that the hardness of cementite Fe_3_C of 10 GPa (~920 HV) is increased with the addition of Cr to 13.5 GPa (20 atom% Cr) [11]. Lentz et al. provided a comparative and comprehensive study of the indentation hardness and indentation modulus of iron-rich borides and carboborides of types Fe_3_(C,B). The Fe_3_(C,B) phase in the Fe-C-B system increased from 11.18 (±0.9) Gpa to 15.94 (±0.72) Gpa. The theoretically predicted hardness values of (Fe,Cr)_3_C phases are in agreement with the experimental measurements reported in the literature.

Using the CALPHAD method, we meticulously studied the elastic constants and their derived properties across the full compositional space. Figure 6 presents the distribution of elastic constants across the entire composition space. The third-order interaction parameters between the (Fe,Cr)_3_(C,B) compositions and elastic constants or elastic moduli were obtained by fitting the composition–property relationship using CALPHAD. Subsequently, we modeled the correlation between the elastic constants and elastic moduli of (Fe,Cr)_3_(C,B) with Cr and B concentrations. This analysis generated a cloud diagram depicting the variations in the mechanical properties of the (Fe,Cr)_3_(C,B) with respect to the components, as shown in Figure 7. The shear modulus and Young’s modulus exhibited similar trends: they decreased with increasing B content in the range of 30–50 at%. In general, the moduli increased with increasing Cr content in the range of 70–80 at% and decreased in the Cr content range of 80–90 at%. Hardness gradually increased with increasing B content with the Cr content of 0.1–15% and 45–50 at%. The maximum hardness, shear modulus, and Young’s modulus were obtained in the Cr content of 70–95 at% and B content of 0–40 at%. The shear modulus and hardness *Hv* were significantly low at a B content of 90–100 at% and a Cr content of zero. It can be inferred that the concentration of Cr predominantly influences the mechanical properties of (Fe,Cr)_3_(C,B). The hardness of the (Fe,Cr)_3_(C,B) phase increased at Cr contents exceeding 70 at% and B contents below 20 at%. Hence, superior mechanical properties can be obtained.

## 4. Conclusions

The highest hardness *Hv*, shear modulus, and Young’s modulus were achieved at Cr and B contents in the range 70–95 at% and 40 at%, respectively. At these concentrations, the material is difficult to compress, resulting in poor machinability. At B contents surpassing 90 at% and at a Cr content of zero, the material exhibited low shear modulus *G* and hardness, resulting in poor resistance to deformation, reduced stiffness, and the ease of plastic processing. In this study, we systematically investigated the phase formation and linear and nonlinear elastic behaviors of M_3_X (M = Fe, Cr; X = C, B) across a multicomponent space. We used a combination of first-principle calculations and CALPHAD models. A composition–phase relationship model was constructed for the of the entire multicomponent space, revealing the influence of the elements on the phase composition. Our theoretical study combining the above models can facilitate accelerated alloy design and can be extended to other multicomponent systems. We performed a high-throughput calculation for a cementite-type (M = Fe, Cr; X = C, B) system, providing a set of practical guidelines to facilitate the efficient designing of wear-resistant material compositions.

## Figures and Tables

**Figure 1 molecules-29-01022-f001:**
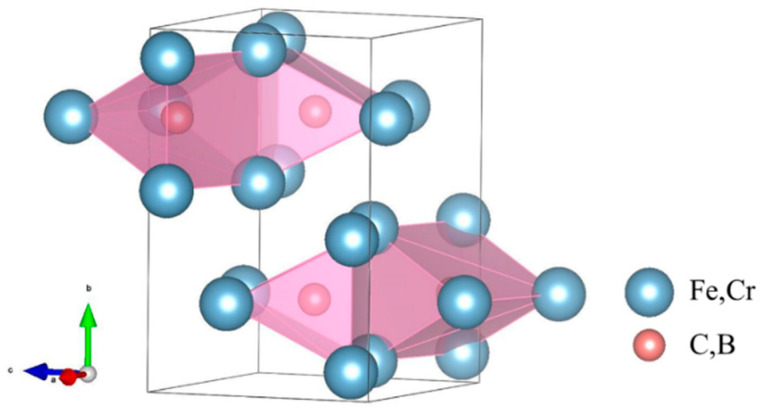
The crystal structure of cementite-type M_3_X shows the orthorhombic space lattice, where blue represents the positions of metal atoms and red represents the positions of non-metal atoms.

**Figure 2 molecules-29-01022-f002:**
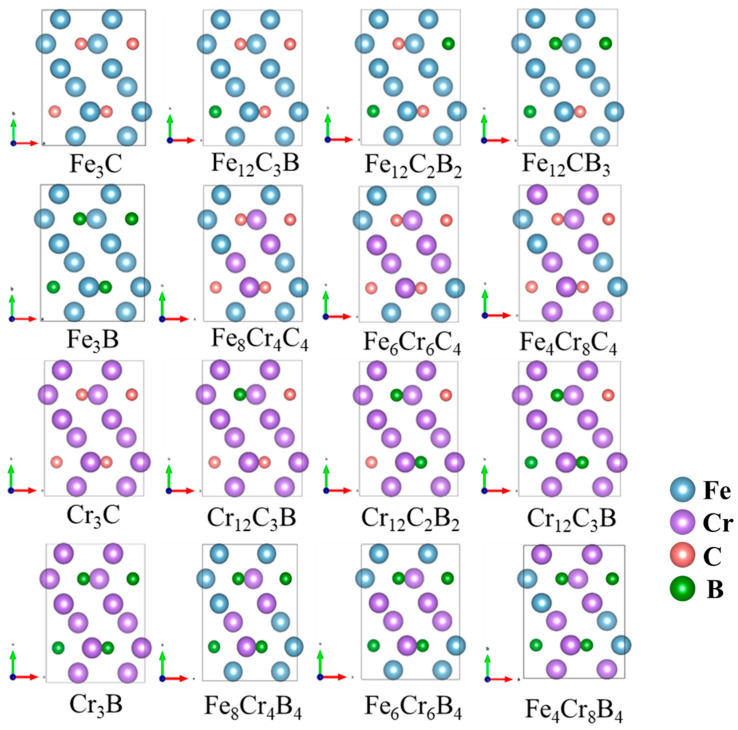
The atomic configurations of Fe, Cr, B and C in a unit cell of (Fe, Cr)_3_(C, B), consisting of formula units (16 atoms). The purple and blue spheres correspond to Fe and Cr atoms, while the red and green spheres represent B and C atoms.

**Figure 3 molecules-29-01022-f003:**
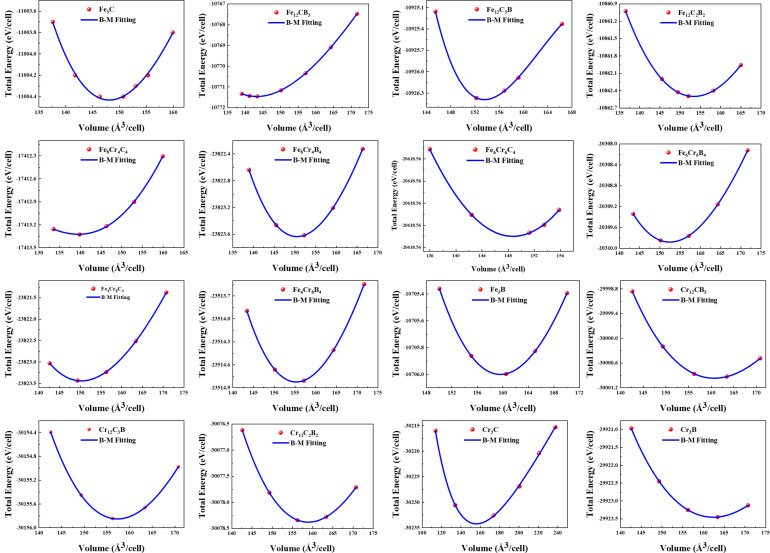
Calculated total energy at zero temperature and without zero point motion as a function of volume of M_3_X (M = Fe, Cr; X = C, B). The filled circles represent calculated points, and the line is a fit to EOS in Equation (1).

**Figure 4 molecules-29-01022-f004:**
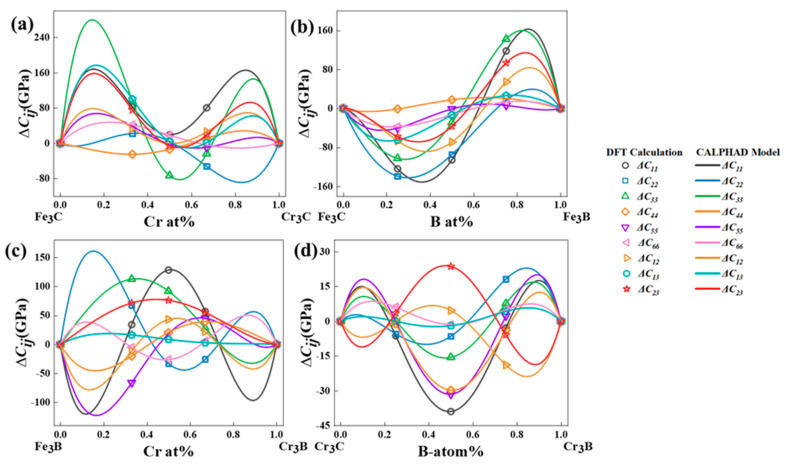
The elastic constants of cementite-type M_3_X (M = Fe, Cr; X = C, B) calculated by DFT and fitted by CALPHAD model. (**a**) (Fe,Cr)_3_C, (**b**) Fe_3_(C,B), (**c**) (Fe,Cr)_3_B and (**d**) Cr_3_(C,B).

**Figure 5 molecules-29-01022-f005:**
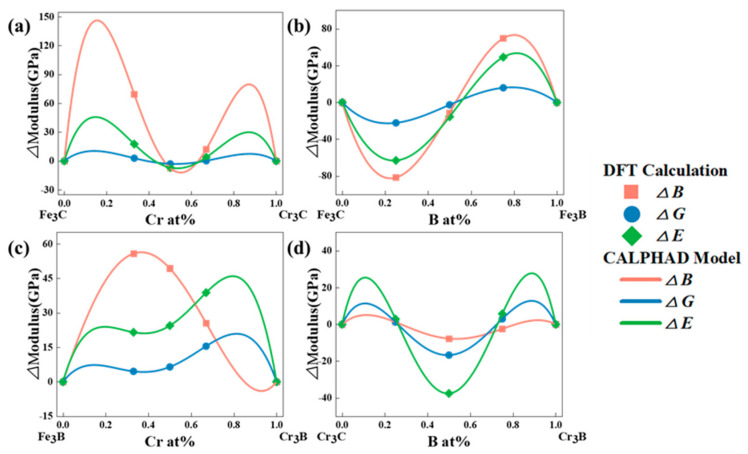
The elastic of modulus cementite-type M_3_X (M = Fe, Cr; X = C, B) calculated by DFT and fitted by CALPHAD model. (**a**) (Fe,Cr)_3_C, (**b**) Fe_3_(C,B), (**c**) (Fe,Cr)_3_B and (**d**) Cr_3_(C,B).

**Figure 6 molecules-29-01022-f006:**
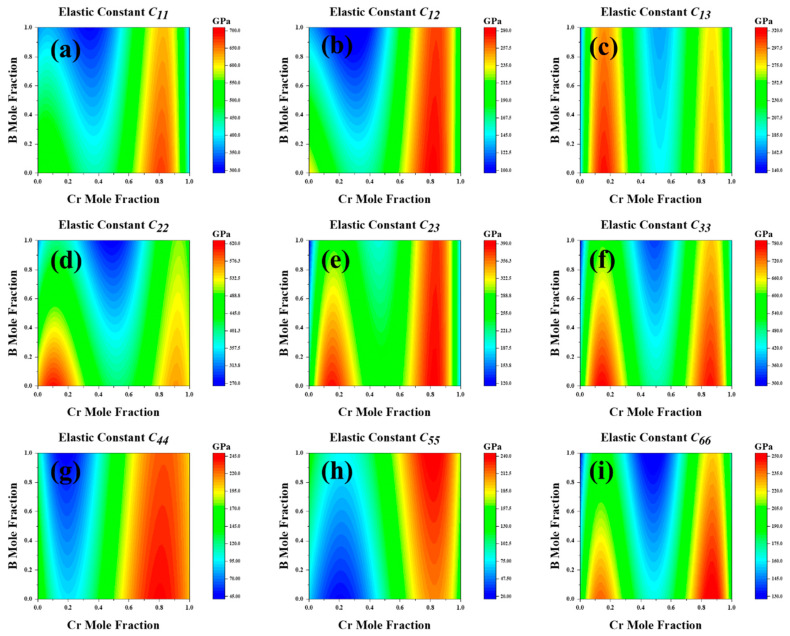
The elastic constants over the whole composition space of cementite-type M_3_X (M = Fe, Cr; X = C, B). (**a**) *C*_11_, (**b**) *C*_12_, (**c**) *C*_13_, (**d**) *C*_22_, (**e**) *C*_23_, (**f**) *C*_33_, (**g**) *C*_44_ (**h**) *C*_55_, (**i**) *C*_66_.

**Figure 7 molecules-29-01022-f007:**
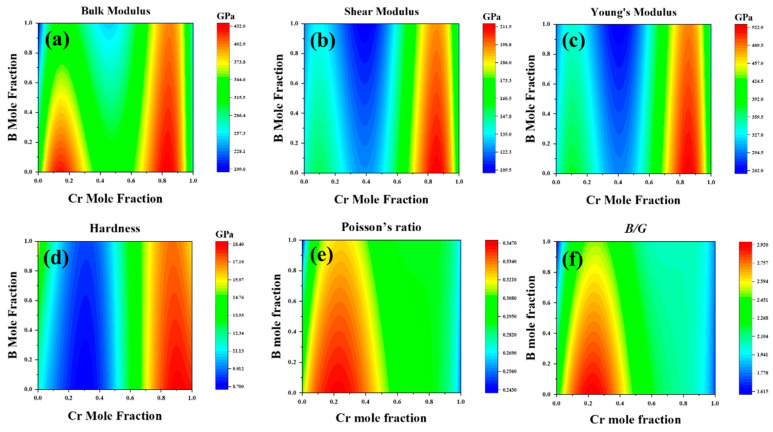
The elastic modulus over the whole composition space of cementite-type M_3_X (M = Fe, Cr; X = C, B). (**a**) the bulk modulus (*B*), (**b**) shear modulus (*G*), (**c**) Young’s modulus (*E*), (**d**) hardness (*Hv*), (**e**) Poisson’s ratio (v), (**f**) *B*/*G*.

**Table 1 molecules-29-01022-t001:** Calculated equilibrium lattice parameters (Å^3^), equilibrium cell volume (Å^3^) and total cell energy (unit in eV/Cell) of cementite-type M_3_X (M = Fe, Cr; X = C, B) at 0 K.

Phases	Lattice Constants (Å)			*V*/Å^3^	*E*_0_ (eV/Cell)	*B*_0_ (GPa)	$B_{0}^{\prime }$
	*a*	*b*	*c*				
Fe_3_C	4.811	6.521	4.31	154.258	−11,013.761	1.984	4.602
Fe_3_C ^a^	5.092	6.748	4.520				
Fe_3_C ^b^	5.062	6.748	4.533				
Fe_8_Cr_4_C_4_	4.884	6.543	4.373	139.612	−17,413.326	1.573	3.803
Fe_6_Cr_6_C_4_	4.967	6.746	4.44	148.753	−20,618.562	2.008	3.203
Fe_4_Cr_8_C_4_	5.119	6.573	4.467	150.277	−23,823.442	1.903	4.945
Cr_3_C	5.191	6.661	4.516	160.851	−30,234.414	1.675	3.923
Cr_3_C ^c^	5.009	6.707	4.456				
Cr_3_C ^d^	5.120	6.800	4.580				
Fe_12_C_3_B	5.273	6.498	4.256	145.852	−10,926.393	1.805	3.827
Fe_12_CB_3_	5.045	6.759	4.504	153.581	−10,771.459	1.983	4.653
Fe_3_B	5.473	6.711	4.387	160.055	−10,706.463	1.425	4.252
Fe_3_B ^e^	5.397	6.648	4.380				
Cr_12_C_3_B	5.198	6.675	4.536	157.377	−30,155.826	1.774	4.269
Cr_12_C_2_B_2_	5.216	6.697	4.551	158.964	−30,078.382	1.734	4.318
Cr_12_CB_3_	5.234	6.721	4.567	160.686	−30,000.972	1.696	4.146
Cr_3_B	5.253	6.744	4.583	162.381	−29,923.457	1.675	3.924
Fe_8_Cr_4_B_4_	5.011	6.714	4.474	150.521	−23,823.636	1.9141	4.905
Fe_6_Cr_6_B_4_	5.352	6.595	4.32	152.474	−20,309.882	1.800	4.499

^a^ Expt. at 298 K: [34]. ^b^ Expt. at 298 K: [35]. ^c^ Expt. at 298 K: [15]. ^d^ Expt. at 298 K: [36]. ^e^ Expt. at 298 K: [37].

**Table 2 molecules-29-01022-t002:** Elastic constants of *C_ij_* from the first principles in the cementite-type M_3_X (M = Fe, Cr; X = C, B) GPa).

Phases	*C* _11_	*C* _22_	*C* _33_	*C* _44_	*C* _55_	*C* _66_	*C* _12_	*C* _13_	*C* _23_
Fe_3_C	383.89	553.09	495.72	174.78	69.11	180.73	236.57	179.14	235.47
Fe_3_B	358.02	323.17	302.63	177.70	132.41	131.72	132.07	155.72	120.11
Cr_3_B	363.43	478.06	415.49	199.81	167.33	167.33	184.81	190.09	175.17
Cr_3_C	552.84	523.00	484.03	205.90	141.68	192.73	176.02	203.86	167.68
Cr_3_C ^a^	518.7	445.6	401.6	193.9	148	202.3	195.3	208.4	212.2
Cr_12_CB_3_	365.67	507.48	440.30	199.06	156.98	161.40	163.76	197.80	167.40
Cr_12_C_2_B_2_	334.73	494.08	434.24	171.43	124.81	167.14	185.09	195.06	195.06
Cr_12_C_3_B	385.06	506.36	467.07	204.31	146.90	186.74	178.32	200.36	173.13
Fe_4_Cr_8_C_4_	517.04	480.47	464.22	186.61	129.33	185.14	225.15	197.31	209.36
Fe_6_Cr_6_C_4_	483.26	534.05	416.59	188.57	91.67	205.01	206.17	195.43	196.12
Fe_12_CB_3_	522.47	397.16	493.26	183.40	137.31	157.18	212.92	188.03	242.54
Fe_12_C_2_B_2_	345.52	343.59	370.37	175.13	118.42	143.31	115.48	154.30	141.29
Fe_12_C_3_B	373.59	356.57	345.10	135.29	83.880	130.90	143.94	108.47	149.03
Fe_8_Cr_4_B_4_	394.02	441.99	452.59	199.04	123.98	138.24	138.24	183.50	209.86
Fe_6_Cr_6_B_4_	489.09	367.81	451.08	206.99	170.29	123.45	201.51	181.61	223.84
Fe_4_Cr_8_B_4_	418.64	401.33	406.39	237.73	195.15	161.14	189.51	181.71	212.32
Fe_8_Cr_4_C_4_	573.47	565.26	587.48	223.45	68.11	226.46	250.64	287.25	288.58

^a^ Cal. at 298 K: [38].

**Table 3 molecules-29-01022-t003:** Fitted parameters of the current CALPHAD model in the cementite-type M_3_X (M = Fe, Cr; X = C, B).

Phases	Llmijn	Δ*C*_11_	Δ*C*_22_	Δ*C*_33_	Δ*C*_44_	Δ*C*_55_	Δ*C*_66_	Δ*C*_12_	Δ*C*_13_	Δ*C*_23_	Δ*G*	Δ*B*	Δ*E*
(Fe,Cr)_3_B	^0^ *L*	−156	−26	−62	−125	−118	−6	19	−8	94.5	−31	−67	−150
	^1^ *L*	18	126	40	13	−15	−6	−101	23	−51	−20	9	15
	^2^ *L*	525	239	332	526	526	140	−274	75	−403	110	312	693
(Fe,Cr)_3_C	^0^ *L*	78	16	−293	−7	−55	73	−1	16	−22	−29	−12	−27
	^1^ *L*	−13	−496	−793	−315	243	−302	−53	−645	−373	−381	−18	−91
	^2^ *L*	2513	−732	3943	636	214	113	1183	1851	2043	1855	165	657
Fe_3_(C,B)	^0^ *L*	−422	−378	−115	−4	71	−51	−275	−53	−146	−45	−11	−62
	^1^ *L*	1290	830	1305	249	116	271	646	487	806	807	203	599
	^2^ *L*	1625	206	888	−343	−73	−54	976	−199	968	51	−26	100
(Fe,Cr)_3_B	^0^ *L*	513	−131	368	73	92	−104	172	35	305	197	26	98
	^1^ *L*	152	−621	−563	739	394	72	222	−90	−108	−201	73	115
	^2^ *L*	−265	1959	−428	−107	−327	909	−126	79	−154	−116	168	335

**Table 4 molecules-29-01022-t004:** Calculated Voigt’s bulk modulus (*Bv*), Reuss’s bulk modulus (*B_R_*), effective bulk modulus (*B*), Voigt’s shear modulus (*Gv*), Reuss’s shear modulus (*G_R_*), effective shear modulus (*G*), Young’s modulus (*E* in Gpa), Poisson’s ratio (*v*) and the universal anisotropic index (*A^U^*) a for cementite-type M_3_X (M = Fe, Cr; X = C, B).

Phases	*B* (GPa)			*G* (GPa)			*E* (GPa)	ν (GPa)	*Hv* (GPa)	*Aᵁ*
	Voigt	Reuss	Hill	Voigt	Reuss	Hill				
Fe_3_C	321.6	319.0	320.3	147.6	129.3	138.5	363.2	0.311	11.643	0.71
Fe_3_C ^a^			237			74				
Fe_3_C ^b^			105			70				
Fe_3_C ^c^			217			69				
Fe_3_C ^d^									10.1	
Fe_3_B	199.9	198.1	199.0	126.7	119.2	123.0	305.9	0.244	16.064	0.32
Fe_3_B ^e^									16.2	
Cr_3_B	261.9	259.3	260.6	149.4	149.4	145.	367.2	0.265	16.037	0.01
Cr_3_C	276.2	273.1	274.6	164.2	155.5	159.9	401.7	0.256	18.076	0.29
Cr_3_C ^f^			291.8			145.6				
Cr_12_CB_3_	263.4	260.3	261.9	155.7	147.7	151.7	381.5	0.257	17.321	0.28
Cr_12_C_2_B_2_	262.4	257.3	259.8	140.2	131.3	135.7	346.9	0.278	14.231	0.36
Cr_12_C_3_B	273.5	271.1	272.3	161.3	153.5	157.4	396.0	0.258	17.730	0.26
Fe_4_Cr_8_C_4_	302.8	301.3	302.0	155.5	150.6	153.1	392.9	0.283	14.970	0.16
Fe_4_Cr_8_C_4_ ^g^			297.1			139.8				
Fe_6_Cr_6_C_4_	292.1	288.3	290.2	152.7	152.7	146.2	375.6	0.284	14.397	0.03
Fe_12_CB_3_	299.9	298.0	299.0	146.8	138.6	142.7	369.6	0.294	13.310	0.30
Fe_12_C_2_B_2_	209.0	207.8	208.4	130.6	125.6	128.1	319.0	0.245	14.437	0.20
Fe_12_C_3_B	208.6	207.9	208.3	114.9	109.8	112.4	285.8	0.271	12.913	0.23
Fe_3_(C,B)-0.2B ^h^									11.18 ± 0.9	
Fe_8_Cr_4_B_4_	275.9	274.2	275.1	138.3	131.3	134.8	347.7	0.289	13.171	0.27
Fe_6_Cr_6_B_4_	280.2	278.1	279.1	146.8	134.1	140.5	361.0	0.284	13.980	0.48
Fe_4_Cr_8_B_4_	265.9	265.7	265.8	161.6	144.9	153.3	385.7	0.258	17.352	0.57
Fe_8_Cr_4_C_4_	375.	374.32	374.89	163.58	133.44	148.5	393.5	0.325	11.070	1.13

^a^ Expt. at 298 K: [47]. ^b^ Expt. at 298 K: [48]. ^c^ Expt. at 298 K: [49]. ^d^ Expt. at 298 K: [50] ^e^ Cal. at 0 K: [51]. ^f^ Cal. at 0 K: [52]. ^g^ Expt. at 298 K: [53]. ^h^ Expt. at 298 K: [11].

## Data Availability

The authors are willing to provide the supporting data of this study upon reasonable request.

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
