# Peer review of "Elastic Properties of Alloyed Cementite M3X (M = Fe, Cr; X = C, B) Phases from First-Principle Calculations and CALPHAD Model"

_molecules, 2024, doi:10.3390/molecules29051022_

Round 1
Reviewer 1 Report
Comments and Suggestions for Authors
The submitted manuscript presents the results of a theoretical study of compounds with the structure of cementite (M3X (M = Fe, Cr; X = C,B)) using first-principle calculations and CALPHAD modeling. Obviously, the authors pay special attention to the volume modulus, shear modulus, Young's modulus, Poisson's ratio and hardness of materials. However, the article should be significantly improved before publication in the journal, according to the following comments:
1. The authors should overview phases of wear-resistant alloys that are relevant for the study (in Introduction).
2. There are several statements in the manuscript that require to be supported with references or experimental data (for example, Page 2, Lines 50-54).
3. Are the configurations under study available in real steels? Are the phases predicted using calculations?
4. In Table 1, the last line is repeated. Please, check and correct.
5. To what extent do the obtained results correspond to the experimental data? Please add the answer to the discussion section
Reviewer 2 Report
Comments and Suggestions for Authors
The manuscript presents new numerical predictions of the bulk moduli, shear moduli, Young’s moduli, Poisson’s ratios of crystal structures of (Fe, Cr)3(C, B) with different concentrations of Cr and B. The computational methods CALPHAD was used for simulation of multicomponent phase behavior. Density functional theory was used for prediction of elastic moduli of crystalline structures. The prediction of the crystalline structures hardness was obtained using the semiempirical equation of hardness proposed by Tian and coworkers (https://doi.org/10.1016/j.ijrmhm.2012.02.021). The phase formation and linear and nonlinear elastic behaviors of M3X (M = Fe, Cr; X = C,B) across a multicomponent space of crystalline structures were studied. It was found that the highest hardness Hv, shear modulus, and Young's modulus were predicted for crystalline structures at Cr and B contents in the range 70–95 at% and 40 at%, respectively. At B contents surpassing 90 at% and at a Cr content of zero, the crystalline structures exhibited low shear modulus G and hardness, resulting in reduced stiffness, and possibility of plastic processing. The manuscript needs serious revision and addition. A significant drawback of the manuscript is the lack of assessment of the reliability of the results of predictions of hardness Hv, shear modulus, and Young's modulus obtained using these methods. In Section 3, the Birch–Murnaghan isothermal equation of state is used for predictions of the bulk modulus. However, the authors do not discuss at what temperature the estimates were made and what values of B0, B0’ were used for different phases. It should be noted that E0 in equation (1) is not the static total energy of the condensed medium volume, as stated in the manuscript, but only the cold component of the internal energy. The manuscript does not contain information about the values of the B0, and B0’ parameter that were used in the calculations for the M3X phases under study (M = Fe, Cr; X = C,B). There is no information on method obtaining numerical values of B0’. In the Murnaghan equation, B0' must be given. It is known that in some situations the value B0’ = 5/3 becomes incompatible with the Thomas–Fermi limit theory. This means that as B0' approaches ~ 5/3, predictions of the mechanical state using the Murnaghan equation become unlikely. In this regard, the methodology for obtaining the results presented in the manuscript should be more strictly justified. Note that the adequacy of the semiempirical equation for prediction of the hardness (11) for the studied crystal systems (Fe, Cr)3(C, B) was not investigated in the cited work by Tian and coworkers [21]. It is necessary to supplement the manuscript with results showing the possibility and degree of adequacy of the semi-empirical relationship used to predict the hardness of the studied crystalline systems M3X (M = Fe, Cr; X = C,B). Attention should be paid to the some discrepancy between the predictions obtained using the formula (11) and the experimental data indicated in the original work of Tian and coworkers. In the absence of such justification, the presented in this manuscript results cannot be considered reliable.В рукописи представлены результаты прогнозов the energy–volume relationship of the cementite M3X (M = Fe, Cr; X = C, B) and the elastic stiffness of M3X (M = Fe, Cr; X = C, B) including three component system Cr12CB3. The authors claim that in 2010, Liu [23] modeled the variations in the mechanical properties of (Fe, Cr)3(C, B) solid solutions with concentration using the CALPHAD method. This statement is not true. Liu and coworkers [23] showed the possibility using CALPHAD method for prediction single-crystal elastic stiffness coefficients for hexagonal closed-packed solution phases in the Mg–Al system and intermetallic phase Al12Mg17. It was shown that calculated the bulk, the shear, and the Young’s moduli have different agreements with experimental data in a wide temperature range. This manuscript deals not only with binary compounds, which also have a different crystal structure than solid solutions in a hexagonal close-packed lattice. It is important, that the CALPHAD method is used really for predictions properties of stable under the considered thermodynamic condition multi-component systems. But the accuracy of forecasts of elastic moduli and stiffness and under arbitrary thermodynamic conditions for the existence of crystalline system phases of M3X (M = Fe, Cr; X = C,B) have to be proved. The manuscript does not contain a comparison of the obtained parameter predictions with experimental data, which could at least partially resolve questions about the calculation methodology

Round 2
Reviewer 1 Report
Comments and Suggestions for Authors
The paper may be accepted in the present form.
Reviewer 2 Report
Comments and Suggestions for Authors
The manuscript presents new numerical predictions of the bulk moduli, shear moduli, Young’s moduli, Poisson’s ratios of crystal structures of (Fe, Cr) 3(C, B) with different concentrations of Cr and B. The computational methods CALPHAD was used for simulation of multicomponent phase behavior. Density functional theory was used for prediction of elastic moduli of crystalline structures. The prediction of the crystalline structures hardness was obtained using the semiempirical equation of hardness proposed by Tian and coworkers (https://doi.org/10.1016/j.ijrmhm.2012.02.021).
The phase formation and linear and nonlinear elastic behaviors of M3X (M = Fe,Cr; X = C,B) across a multicomponent space of crystalline structures were studied.
It was found that the highest hardness Hv, shear modulus, and Young's modulus were predicted for crystalline structures at Cr and B contents in the range 70–95 at% and 40 at%, respectively. At B contents surpassing 90 at% and at a Cr content of zero, the crystalline structures exhibited low shear modulus G and hardness, resulting in reduced stiffness, and possibility of plastic processing.
After revisions and additions taking into account the recommendations of the reviewers, the quality of the manuscript improved significantly.
The manuscript of the article is well structured. The list of references is satisfactory. The conclusions are based on the analysis of the obtained results.
The results presented in the manuscript may be of interest to a wide range of specialists, graduate students and students who study Fe-Cr-C-B wear-resistant steels.